# The Gluopsins: Opsins without the Retinal Binding Lysine

**DOI:** 10.3390/cells11152441

**Published:** 2022-08-06

**Authors:** Martin Gühmann, Megan L. Porter, Michael J. Bok

**Affiliations:** 1School of Biological Sciences, University of Bristol, Bristol BS8 1TQ, UK; 2Department of Biology, University of Hawai’i at Mānoa, Honolulu, HI 96822, USA; 3Lund Vision Group, Department of Biology, University of Lund, 223 62 Lund, Sweden

**Keywords:** opsin, evolution, photoisomerase, phylogeny, G-protein-coupled receptor, chemoreceptor, data mining, retinal binding site

## Abstract

Opsins allow us to see. They are G-protein-coupled receptors and bind as ligand retinal, which is bound covalently to a lysine in the seventh transmembrane domain. This makes opsins light-sensitive. The lysine is so conserved that it is used to define a sequence as an opsin and thus phylogenetic opsin reconstructions discard any sequence without it. However, recently, opsins were found that function not only as photoreceptors but also as chemoreceptors. For chemoreception, the lysine is not needed. Therefore, we wondered: Do opsins exists that have lost this lysine during evolution? To find such opsins, we built an automatic pipeline for reconstructing a large-scale opsin phylogeny. The pipeline compiles and aligns sequences from public sources, reconstructs the phylogeny, prunes rogue sequences, and visualizes the resulting tree. Our final opsin phylogeny is the largest to date with 4956 opsins. Among them is a clade of 33 opsins that have the lysine replaced by glutamic acid. Thus, we call them gluopsins. The gluopsins are mainly dragonfly and butterfly opsins, closely related to the RGR-opsins and the retinochromes. Like those, they have a derived NPxxY motif. However, what their particular function is, remains to be seen.

## 1. Introduction

Opsins are the molecules that allow us to see. They are G-protein-coupled receptors (GPCRs) [1,2], which are chemoreceptors and have seven transmembrane domains forming a binding pocket for a ligand [3,4]. The ligand for opsins is 11-cis-retinal [5,6,7,8,9], which is covalently bound to a lysine residue [10] in the seventh transmembrane domain [11,12,13]. However, 11-cis-retinal only blocks the binding pocket and does not activate the opsin. The opsin is only activated when 11-cis-retinal absorbs a photon of light and isomerizes to all-trans-retinal [14,15], the receptor activating form [16,17]. Thus, a chemoreceptor is converted to a light or photo(n)receptor.

Opsins and other GPCRs have a number of conserved sequence motifs and residues that are functionally important. All GPCRs have a highly conserved NPxxY^7.53^ sequence motif in their seventh transmembrane domain. Here, we use the common GPCR numbering scheme for residues from Ballesteros and Weinstein [18]. The number before the period is the number of the transmembrane domain. The number after the period is set arbitrarily to 50 for the most conserved residue in that transmembrane domain among GPCRs known in 1995. For the seventh transmembrane domain, the proline in the NPxxY^7.53^ motif is P^7.50^, the asparagine before is then N^7.49^, and the tyrosine three residues after is then Y^7.53^. The NPxxY^7.53^ motif is important for G-protein activation. For instance, if proline^7.50^ is replaced by alanine^7.50^ then cattle rhodopsin has 141% of wild type activity [19], but this depends on the receptor. Other GPCRs have less activity [20] or have no activity at all [21]. However, a receptor could also have activity with alanine^7.50^ and lose it with proline^7.50^ [22]. The same is true for asparagine^7.49^ [19,23] and tyrosine^7.53^ [19,24].

The lysine in the seventh transmembrane domain of cattle rhodopsin (*Bos taurus*) is the 296th amino acid [12,25] and thus is named lysine 296^7.43^ (here we also include the Ballesteros and Weinstein numbering). Cattle rhodopsin was the first opsin whose amino acid sequence was determined [25]. Since lysine 296^7.43^ binds retinal, opsins without it are not photosensitive [26]. Other opsins may have more or fewer amino acids than cattle rhodopsin and the corresponding lysine may be at another position. However, the corresponding lysine can be easily identified by aligning those opsins to cattle rhodopsin. For simplicity, we call such homologous lysines also lysine 296^7.43^ in accordance with the opsin literature.

Lysine 296^7.43^ is well conserved among opsins, so well conserved that sequences without it are not even considered opsins and thus excluded from large scale phylogenetic reconstructions [27,28]. Feuda et al. [29] even reconstructed a group of *Trichoplax* GPCRs without lysine 296^7.43^ that they found closely related to the opsins and thus called placopsins.

Beside light detection, some opsins are also involved in thermosensation [30], mechanoreception such as hearing [31] and other functions [32,33]. Recently, opsins have even been identified that can act as aristolochic acid chemoreceptors, even if light sensitivity is abolished by replacing lysine 296^7.43^ by another amino acid [26]. These studies suggest a functional flexibility in opsins to facilitate tasks beyond photoreception. Therefore, we wondered: Do opsins exist that have lost lysine 296^7.43^ during evolution? Such opsins would be nested within other groups of opsins that still have lysine 296^7.43^. Here, we built a new custom pipeline for reconstructing phylogenies and reconstructed the largest opsin phylogeny to date. In this phylogeny, we found a clade of opsins that have lost lysine 296^7.43^, which we call gluopsins.

## 2. Material and Methods

### 2.1. Protein Sequence Collection

To collect GPCR-protein sequences, we searched with BLAST [34] in the uniprot databases SPROT and TREMBL locally and in the NCBI databases nr, refseq_protein, swissprot, and tsr_nr remotely for sequences similar to opsins. As opsin bait sequences, we used 84 sequences from the data set of Ramirez et al. [35]. The bait sequences were spread over their phylogeny to cover a broad range of opsins. The chosen sequences are in Appendix A; some bait sequences turned out to be rogue, which are in Appendix A (see Section 2.2). We used a liberal e-value cutoff of 1 × 10^−5^ and collected the first 100,000 hits. Among the bait sequences, we used also a placopsin sequence. We restricted the number of queries to the servers of NCBI to one sequence per database at the same time so that we would not overuse this common and public resource. Additionally, we added sequences from Lowe et al. [36] and D’Aniello et al. [37], which were also used by Ramirez et al., but were not in one of the databases we searched.

We also added transcriptome sequences from the marine worm *Platynereis dumerilii* [38], which are available at https://jekelylab.ex.ac.uk/blastdbs/index.html (accessed on 1 June 2022) as assembly version 2. These sequences were tentatively annotated as opsins via BLAST. We also added sequences from fan worm transcriptomes that we identified as opsin related with our own version of Phylogenetically Informed Annotation (PIA, [39]). (The new sequences and all other sequences of our final tree are included in Appendix A). Our version of PIA is derived from that of Pérez-Moreno et al. [40] and is available at https://github.com/MartinGuehmann/PIA2 (accessed on 1 June 2022).

### 2.2. Sequence Pruning

Since we collected sequences from different databases, we collected duplicates, which we removed with SeqKit [41]. However, the dataset still contained many very similar sequences. Therefore, we grouped the sequences that were 90% or more similar to each other into clusters and chose from each a representative with CD-Hit [42,43]. This data set of representatives contained 89,996 sequences. However, since our sequence search was very permissive, many of those were non-opsin sequences. Using so many sequences to reconstruct a phylogenetic tree consumed more time and memory than our cluster computer could provide.

To purge non-opsin sequences, we split the dataset randomly into 128 subsets of about 900 sequences with SeqKit and added our opsin bait sequences to each set and a non-opsin sequence of an olfactory receptor. Then, we aligned each subset with PASTA [44] and reconstructed for each subset a phylogenetic tree with IQ-Tree 2 [45]. Each tree was rooted with nw_reroot at the olfactory receptor so that we could extract the subtree spanned by our bait sequences with nw_clade from the newick utilities [46]. For this, we had to remove seven sequences from the set of bait sequences including the placopsin and Go-opsin2 of *Terebratalia transversa*, since those were placed in some trees outside the opsins and thus would have given us non-opsin sequences.

From the trees, we extracted 8483 potential opsin sequences. We added back 1000 sequences that were randomly chosen by SeqKit to include a diverse non-opsin outgroup. Since we were interested in the placopsins too, which were removed because of the placement problem of the placopsin bait sequence, we also added back all sequences from *Trichoplax*. This way, our final dataset contained 9694 sequences before rogue removal.

### 2.3. Rogue and Long Branch Removal

Rogue sequences introduce instability to a phylogeny by jumping around from one place to another and may change the relationships and branch supports within the phylogeny unpredictably [47,48]. To remove such rogue sequences as well as long branches, we split the dataset randomly into subsets of about 900 sequences, aligned each subset with PASTA and built a phylogenetic tree and bootstrap trees with IQ-Tree 2. The bootstrap trees were passed to RogueNaRok [47,49] to identify rogue sequences and the main consensus tree was passed to TreeShrink [50] to identify long branches. The rogue sequences and the sequences of the long branches were removed from the main dataset with SeqKit. However, since what a rogue sequence is, depends also on the other sequences in the dataset [48], we repeated this splitting and removal procedure for 20 iterations. Once the 20 iterations were complete, we used iteration 10 as a base and built from there trees containing all sequences up to iteration 20. In each iteration, we removed the sequences identified as rogue sequences from the full tree and from the previous split trees at that iteration to build the next full tree. Our final data set at iteration 20 contained 6040 sequences. The sequences are available in Appendix A.

### 2.4. Phylogenetic Reconstruction

For phylogenetic reconstructions, we aligned the sequences with PASTA [44] with the default settings. From the alignment, we removed columns that contained more than 90% of sequences with a gap with TrimAl [51]. The trimmed alignment was then passed to IQ-Tree 2. IQ-Tree 2 selected the best substitution model (JTT + F + G4 for our final tree) for inferring a maximum likelihood tree and generated three kinds of support values: Shimodaira–Hasegawa-like approximate likelihood ratio test (SH-aLRT) values [52], aBayes [53], and ultra-fast bootstrap (UFBoot) values [54,55]. For both the SH-aLRT and the UFBoot values, we used 1000 replicates.

We used more than one kind of support value, because one may support a wrong branch while the other may not so that they may “compensate for each other’s failures” [52]. We rejected a branch if its SH-aLRT value was below 0.1 irrespective of whatever the other support values were [56] and accepted it if all three support values were above or equal to the following thresholds: 80 for SH-aLRT [52], 0.95 for aBayes [53], and 95 for UFBoot [54].

For alignment trimming, we used TrimAl with a moderate trimming value. In principle, removing data such as N- and C-terminal sequences of opsins, as done previously [28], can also remove phylogenetically informative sites and thus reduce the resolution of the phylogenetic reconstruction [57]. In fact, TrimAl did not improve the averages of support values as we could see, it may even have slightly worsened them. This agrees with rigorous benchmark tests with real and simulated data [58,59,60]. However, since the aligner introduced many gaps with columns almost empty, TrimAl reduced the alignment file size: For instance, the alignment with all sequences originally had a size of 1 GB and after trimming had a size of 11 MB. This significantly reduced the time IQ-Tree needed to reconstruct the phylogeny. The final gap-reduced alignment is available in Appendix A, a version of the alignment sorted by the order of sequences in the final tree is available in Appendix A, and the final tree is available in newick format in Appendix A.

### 2.5. Tree Visualization

The trees were visualized with ETE 3 [61], which is a python package for programmatically visualizing phylogenetic trees. For that, we used a CSV file to define each clade with a name, a color to be used within a tree, and a sequence ID. The sequence ID represents a clade such as the peropsins and points to a leaf in the tree.

To annotate the tree, we rooted the master tree at the outgroup leaf, which is defined as the last entry in the CSV file. Then, from each clade leaf, we traversed to the root and counted for each visited node how many clade leaves were descendants. The descendant count is the highest at the root and stays that high when traversed to any clade defining leaf back as long as all these leaves are decedents. This way, we could determine the last common ancestor of all ingroup clades and used that to reroot the tree. Similarly, we could define the root node of each clade, which is the last node from the defining leave node with the descendant count of one. From that node we could color the clade or collapse the clade. We saved each clade as an independent tree.

This way, we did not only visualize the complete tree, but also the trees from the partial datasets we used for the initial rogue removal so that we could inspect them better. However, in these reduced trees, the sequence we used to define the clade leaf may not be included. Therefore, we used the clade subtrees from the main tree to find an alternative clade leaf, which we then could use to collapse or color the clades accordingly.

With ETE 3, we produced for each tree a pdf file with all branches, and a tree collapsed at the clade root nodes. We also printed the support values onto in the order SH-aLRT/aBayes/UFBoot the branches. In the collapsed tree, each value is represented by a pie chart. The pie chart is black for the following values SH-aLRT ≥ 80%, aBayes ≥ 0.95, and UFBoot ≥ 95%; otherwise, it is gray.

### 2.6. Finding Position 296^7.43^ in Other Opsins and GPCRs

To find opsins that have lost lysine 296^7.43^, we used the gap-reduced alignment to determine the amino acid position in the other opsins and GPCRs that corresponds to lysine 296^7.43^ in cattle rhodopsin. Since this alignment was gap reduced, we aligned the reference sequence of cattle rhodopsin to the gap-reduced cattle rhodopsin in the alignment. This allowed us to map lysine 296^7.43^ to its actual position in the gap reduced alignment, and we could get the corresponding amino acid at the corresponding position for all sequences. This information was then applied to the trees generated by ETE 3. For the collapsed trees, this is simply the percentage of each amino acid of the sequences at that position in the collapsed subtree.

We did not consider single opsins without lysine 296^7.43^ isolated within a clade as real, because we could not exclude that those were missequenced, misassembled, or pseudogenes.

### 2.7. Annotate the Sequences with Higher Taxa

We annotated the sequences in the tree with information about the corresponding higher taxa. For that, we used the NCBI taxonomy database, which our pipeline downloaded automatically and extracted all the information about the known genera. The pipeline checked the sequence IDs in the tree for whether they contained a genus string from the database. If it was in the database, the corresponding higher taxon was assigned to that sequence and if it was in our list of interesting taxa. We started with the taxon list of Porter et al. [28], and checked unidentified sequences, whether they were from a genus that was not within a higher taxon on our list of interesting taxa. In that case, we added that higher taxon. Some of these unidentified sequences had a sequence ID that did not contain a genus name, and thus we could not annotate it with a higher taxon. However, since this was only the case for 26 of 6040 sequences, we do not consider this as a problem.

### 2.8. Sequence Logo

For each defined clade, we generated a sequence logo around lysine 296^7.43^ with Logomaker [62] and the library matplotlib [63]. The logo starts at residue 287^7.34^, ends at residue 324^7.71^, and spans 37 residues. This region does not contain any gaps in most gluopsins and contains conserved residues. Beside residue 296^7.43^, we highlighted the residues of the NPxxY^7.53^ motif, and the residues 292^7.39^ and 314^7.64^.

### 2.9. The Phylogeny Pipeline

Finally, we combined all the steps together with standard Linux tools including Bash. This way we built a pipeline with the aim of putting raw data in and getting publication ready figures out, while everything in between was processed, automatically (Figure 1).

Since this is a resource-intensive pipeline, we built it for the cluster computer BluePebble at the University of Bristol. BluePebble had computer nodes with up to 187 GB memory and 24 processor cores, and a maximum wall time of 72 h. Its scheduler system was first PBS-Pro, which was later replaced by Slurm. Therefore, the pipeline can be used with either PBS-Pro or Slurm, and if necessary other schedulers might be easily added. The code was modularized such that jobs could execute tasks in parallel or in sequence when the task depended on the result of a previous task.

The pipeline has two input types, the bait sequences, and the clade definitions. The bait sequences are used to find similar sequences in the public databases with BLAST. They are also used to filter for opsin sequences. However, for that, they need to be indeed opsins. However, some bait sequences turned out not to be opsins or to be rogue sequences. Therefore, the pipeline can be paused at the filter step and these sequences can be marked as additional bait sequences to keep them for a total rerun, but not for filtering. We checked in the filter step by checking the trees generated by that step whether the bait sequences defined a clade that only contained opsins. For that, we used Dendroscope 3 [64].

The clade definitions were used for tree visualization and need also be updated after the first run. One reason was that CD-Hit clusters the sequences into groups of 90% similarity, and chooses one sequence as representative, but which one was beyond our control. The other reason was that the sequence that was supposed to define the clade was removed. This was the final and least computational step, which we ran on a Linux laptop after copying the data from the cluster computer, because ETE3 [61] required for tree visualization QT5, which required a running x-server, which is not available on a cluster computer in automatic non-interactive mode.

The code for the phylogeny pipeline is available at https://github.com/MartinGuehmann/PhylogenyPipeline (accessed on 1 June 2022) The data and the files to run it are available at https://github.com/MartinGuehmann/Opsins (accessed on 1 June 2022).

### 2.10. Opsin Nomenclature

During phylogenetic reconstruction, we encountered both new and previously unnamed clades of opsins. We aimed to give them names that strike a balance between descriptive content, uniqueness, and brevity. Our names typically consist of an opsin suffix and a presyllable derived from either a name from a species or higher taxon within the clade, or from a shared property of the opsins in the clade.

## 3. Results

### 3.1. The Phylogeny Pipeline

We reconstructed the relationship of the opsins to find a clade of opsins that has lost lysine 296^7.43^. For that, we used our custom phylogeny pipeline, which automatically executes almost all steps from sequence collection to phylogenetic reconstruction. However, when we filtered the collected sequences for opsins, a few of the original bait sequences turned out to be rogue sequences, grouping sometimes with non-opsin sequences. Furthermore, we could not recover the placopsins as a sister group of the opsins as we expected from the results of Feuda et al. [29]. Therefore, we removed the placopsins and the rogue sequences in the opsin filtering step.

The whole pipeline collects the sequences from the databases; makes the sequences unique within 90% of sequence identity; removes non-opsin sequences, except (in our case) the Trichoplax sequences and adds 1000 randomly chosen outgroup sequences; removes rouge sequences; and builds the final opsin tree, which then only needs to be visualized on a computer running an X-server (Figure 1).

Thus, the entire phylogenetic reconstruction can be reproduced easily. The pipeline can also be used for other GPCRs or proteins, by inputting different bait sequences and clade definitions.

### 3.2. Five Basal Types of Opsins

To find opsins that have lost lysine 296^7.43^, we collected as many sequences as possible. We ended up with 89,996 unique GPCR sequences. Using so many sequences to reconstruct a phylogeny was computationally not feasible for us. Therefore, we automatically removed all non-opsin sequences and added back 1000 randomly chosen sequences plus 211 *Trichoplax* GPCRs as an outgroup. This gave us a reduced dataset of 9694 sequences. Additionally, we increased the quality of the phylogenetic reconstruction by repeatedly removing rogue sequences with RogueNaRok [47,49] and long branches with TreeShrink [50]. After rogue and long branch removal, our final phylogeny contained 6040 GPCRs including 4956 opsins (Figure 2D).

With this, we not only found the gluopsins, the opsins that have lost lysine 296^7.43^ during evolution, but also reconstructed the largest yet opsin phylogeny with 4956 opsin sequences. This phylogeny is more than five times bigger than previous opsin phylogenies [28,35]. This 5k opsin phylogeny is itself interesting, but out of scope here. Therefore, we will describe it elsewhere so that we can focus on the gluopsins, here.

In addition to the 4956 opsins, we recovered an outgroup of 1084 sequences, which apart from two sequences do not have lysine 296^7.43^, but a variety of other residues (Figure 2A and Appendix A). In contrast, most opsins sequences that are full length have lysine 296^7.43^ (Figure 2A–C). This means if our sample of non-opsin GPCRs is representative, then a lysine 296^7.43^ still indicates that a GPCR is most likely an opsin.

We did not recover any bathyopsins or chaopsins as reconstructed by Ramirez et al. [35]. Their bathyopsins and chaopsins have only four and seven sequences, respectively, which are not in our final data set and thus were removed by RogueNaRok. The same happened to the ctenopsins, which is expected as they are known to behave like rogue sequences; in phylogenies, they jump around depending on the outgroup used [65].

Our phylogeny recovered five primary opsin clades: the ciliary opsins (cilopsins), the rhabdomeric opsins (rhabopsins), the tetraopsins, the xenopsins, which were first reconstructed by Ramirez et al. [35], and the nessopsins (Figure 2A). The cilopsins and the rhabopsins do not contain sequences from cnidarians (Figure 2A) but contain the visual opsins of vertebrates and arthropods, respectively. The xenopsins are absent from deuterostomes and the only clade with cnidarian opsins beside the nessopsins (Figure 2A).

The nessopsins contain mainly cnidarian opsins (Figure 2A). They are identical to an unnamed group that fell sister to the cilopsins, the rhabopsins, and the bathyopsins in the phylogenetic tree of Ramirez et al. [35]. They are also identical with the anthozoan opsins II of Quiroga Artigas et al. [66] and the cnidarian opsins of Rawlinson et al. [67]. We identified these groups as nessopsins, since they share the sequence XP_015773304 of *Acropora digitifera* with our nessopsins. Since the nessopsins have had so far no established name, we call them nessopsins after the German word “Nesseltiere”, which means cnidarians.

The tetraopsins are the main group that contain the gluopsins. They are also known as RGR/Go-opsins or Group 4 opsins [28,29,35]. The tetraopsins, like the cilopsins and the rhabopsins, do not have cnidarian sequences (Figure 2A) and are subdivided into three clades: the neuropsins, the Go-opsins, and the chromopsins (Figure 2B).

### 3.3. The Chromopsins

Classically, the chromopsins contain the peropsins, RGR-opsins, and retinochromes (Figure 2C). We derived the name chromopsin from the retinochromes, because they were discovered first [68,69], before the RGR-opsins [70,71] and the peropsins [72]. Additionally, we reconstructed four more clades: the varropsins, astropsins, nemopsins, and gluopsins. How these clades relate to each other is unclear due to low support values (Figure 2C). Two chromopsin orthologs exist in deuterostomes, and one in protostomes.

The peropsins contain sequences from craniates and cephalochordates (Figure 2C). They exclude protostome sequences that were previously described as peropsins [73]. Instead, we reconstructed these sequences, either as retinochromes, gluopsins, or varropsins.

The varropsins only exist in *Limulus* and the arachnids, which are both arthropods (Figure 2C, Appendix A). We named them after *Varroa destructor*, a mite with a varropsin. Although, varropsins have been phylogenetically described as peropsins, their relationships to vertebrate peropsins are unclear due to low support values [73,74,75]. Henze and Oakley [73] actually distinguish between two peropsin clades: the insect and non-insect arthropod peropsins, which are our gluopsins and varropsins, respectively. Their peropsin clades are even interspersed by a sequence from the marine ragworm *Platynereis dumerilii*, which was originally described as a peropsin [76] and has been reclassified as a retinochrome [35]. We do not have enough support to place the varropsins confidently either. Therefore, they could simply be arthropod peropsins or be indeed a different clade.

The RGR-opsins (short for Retinal G protein coupled receptors) have sequences from craniates, hemichordates, and echinoderms (Figure 2C, Appendix A), while the retinochromes have sequences from mollusks, platyhelminths, and annelids (Figure 2C). The annelid sequences come from our transcriptomes except one sequence from *Platynereis dumerilii*, which was originally describes as a peropsin [76], and later reclassified as a retinochrome [35] in agreement with our phylogeny.

The astropsins are echinoderm specific chromopsins (Figure 2C). We named them after *Asterias rubens*, a sea star with an astropsin. Only three astropsins remain in our rogue-pruned final tree, only two cover the seventh transmembrane domain, and one has lysine 296^7.43^ replaced by a glutamic acid (Figure 2C). With only two sequences, it is hard to draw conclusions. However, in total three astropsins with glutamic acid 296^7.43^ have been reported [37]. We checked our alignments from previous iterations of pruning and found three sequences with glutamic acid 296^7.43^ (data not shown). These sequences are all from sea urchins, while sea stars and sea cucumbers have astropsins with lysine 296^7.43^.

### 3.4. The Nemopsins Have Arginine at Position 296

The nemopsins are nematode chromopsins. Only two nemopsins remain in our final tree, and only one covers the seventh transmembrane domain where lysine 296^7.43^ is replaced by an arginine (Figure 2C, Figure 3, Appendix A). Among the removed sequences are sequences from *Caenorhabditis elegans* (NP_001364737.1) and *Pristionchus pacificus* (PDM61246.1); both species are model systems and both sequences have the arginine. The sequence from *C. elegans* has been previously described as an opsin like GPCR with arginine 296^7.43^ and a conserved NPxxY^7.53^ motif. It was named sro-1, short for serpentine receptor class o 1 and is expressed in chemosensory cells [77]. The nemopsins have not been included in a previous opsin phylogeny, and we are not aware of anything else known about them.

### 3.5. The Gluopsins Have Glutamic Acid at Position 296

The gluopsins are arthropod chromopsins mainly specific to butterflies and dragonflies (Figure 2C, Appendix A). Their lysine 296^7.43^ is replaced by a glutamic acid residue (Figure 2C and Figure 3). Therefore, we call them gluopsins, where glu is the three-letter abbreviation of glutamic acid. Our gluopsins form a clade of 36 opsins with 33 gluopsins having glutamic acid 296^7.43^, two are fragments without the seventh transmembrane domain or parts of it, and one is misassembled (Appendix A: Sorted alignment). However, since we have 33 gluopsins with glutamic acid and since we have them from different higher insect taxa, we can exclude that those are missequenced, misassembled or pseudogenes. With glutamic acid 296^7.43^, the gluopsins (and the astropsins and nemopsins) are special since all other opsins have lysine at this position. Interestingly, gluopsins and astropsins have apparently evolved glutamic acid 296^7.43^ independently (Figure 2C).

In contrast to the astropsins, the gluopsins are better studied. Gluopsins in the dragonflies *Sympetrum frequens* (in our tree: BAQ54696.1, Appendix A) and *Orthetrum albistylim* are expressed sparsely in visual organs of the larva and the adult. These gluopsins have also been experimentally verified by reverse transcriptase PCR and sequencing. Despite this, the original study did not mention glutamic acid 296^7.43^ [78]. Our dataset also contains sequences from butterflies and moths such as the common silk moth (*Bombyx mori*), which is of commercial interest, and the tobacco hawk moth (*Manduca sexta*) (XP_021206870, XP_030031533, respectively). Both species are model systems.

Previous opsin phylogenies did not include the gluopsins or notice their glutamic acid 296^7.43^. The gluopsins are neither in the datasets of Porter et al. [28] nor of Ramirez et al. [35], because any sequences without lysine 296^7.43^ (apart from some outgroup sequences) were excluded. The gluopsins were previously reconstructed as insect peropsins by Henze and Oakley [73], who however did mention glutamic acid 296^7.43^. Böhm et al. [79] described two putative gluopsins as peropsins, which came from head transcriptomes of scorpionflies. They did not mention glutamic acid 296^7.43^, either. Even so one sequence has it in their alignment while the other is a fragment without the seventh transmembrane domain. These sequences seem not to have been submitted to a sequence database and thus are not in our phylogeny. Böhm et al. [79] also mentioned a sequence from the jewel beetle *Agrilus planipennis* (XP_025829857), which also has glutamic acid 296^7.43^. This sequence is probably a gluopsin too, even so it was removed from our dataset as a rogue sequence with other sequences from planthoppers, whiteflies, and termites, which all have glutamic acid 296^7.43^ and a derived NPxxY^7.53^ motif.

### 3.6. The NPxxY^7.53^ Motif Is Derived in Some Chromopsins

The NPxxY^7.53^ motif in GPCRs is important for G-protein binding and signaling, and thus is conserved. However, the motif has mutations in RGR-opsins and retinochromes. Thus, these RGR-opsins and retinochromes have been claimed not to signal but to work as photoisomerases instead [80,81]. Therefore, we checked whether the varropsins, the astropsins, and the gluopsins also have mutations in their NPxxY^7.53^ motif, as this could give us some clues about their function. To answer this question and to check for other conserved residues, we generated a sequence logo for each clade of the chromopsins (Figure 3).

The chromopsins have additional conserved residues such as proline 292^7.39^ and arginine 314^7.64^, which is even shared with the other opsins. Besides that, the chromopsins fall into two groups: One group has the peropsins, varropsins, and nemopsins with a well conserved NPxxY^7.53^ motif and the other group has the other chromopsins clades. Interestingly, each clade has its own mutations in the NPxxY^7.53^ motif: The RGR-opsins have NAxxY^7.53^, while the retinochromes VPxxY^7.53^ for annelids or YPxxY^7.53^ for mollusks (Figure 3, Appendix A: Sorted alignment). The astropsins also have mutations within the motif, but since the sequence logo only contains two sequences, it is hard to say what the consensus is. Finally, the gluopsins have the most derived motif of all chromopsins. Most have either PVxxY^7.53^ or PLxxY^7.53^ (Figure 3). The sequences from Böhm et al. [79] and the jewel beetle (XP_025829857) also have PVxxY^7.53^.

Whether these two groups are also phylogenetic groups, is unclear due to low support values. All chromopsins with a derived NPxxY^7.53^ motif have different mutations, which means they could have evolved independently. However, this could still hint to shared functional requirements, which include also relaxed requirements if these chromopsins do not signal.

## 4. Discussion

### 4.1. The Number of Chromopsins in the Urbilaterian

The retinochromes, like the RGR-opsins and the gluopsins of the chromopsins, have a derived NPxxY^7.53^ motif, while the motif of the nemopsins, the peropsins, and the varropsins is conserved (Figure 3). Therefore, we could assume that also the RGR-opsins, retinochromes, and the gluopsins form a group. However, our reconstruction does not recover that relationship with certainty, and the mutations, which are different for the three groups in the motif, suggest independent evolution. In principle based on our final pruned dataset, the urbilaterian could have had one paralogue of the chromopsins that was then duplicated in the craniate lineage. The protostomes have only one paralogue of either retinochromes, varropsins for arachnids, or gluopsins for beetles, scorpionflies, butterflies, and dragonflies. Possibly, the gluopsins are specific to insects, and have been lost in some clades.

For the gluopsins, we could find more insect taxa in our original dataset such as planthoppers, whiteflies, and termites. These sequences have glutamic acid 296^7.43^ and a derived NPxxY^7.53^ motif, which we can use as a diagnostic feature, to identify them as gluopsins. We also built a tree from the original dataset, where some sequences of annelids and mollusks were reconstructed as true peropsins (data not shown). However, we should be careful here, because the peropsins do not have such clear features as the gluopsins and the point of pruning rogue sequences was to reconstruct a more reliable phylogeny. In the end, we cannot exclude that the urbilaterian had as many chromopsin paralogues as the seven we have reconstructed.

Postulating so many chromopsins that have been gained in the urbilaterian and then have been lost in the decedents is an unparsimonious hypothesis. In fact, the general assumption is that gains and losses of genes are rare, and therefore reconciliations of gene trees with species trees try to reduce such gains and losses [82,83].

However, opsin gains and losses are common among vertebrates [84] and are also known in protostomes. The water flea [85], the pineal shrimp [86], dragon and damselflies [78,87], *Limulus* [88], and the mantis shrimp [89] have all gained opsins. In contrast, *Drosophila melanogaster* has only seven opsins, which are all rhabopsins [90]. Therefore, it must have lost the xenopsins, cilopsins, and the three paralogues of the tetraopsins since it evolved from the urbilaterian, as those are present in protostomes and deuterostomes. These are at least five opsin classes we show expanded in our tree (Figure 2). Furthermore, Ramirez et al. [35] concluded that the last common ancestor of deuterostomes and the protostomes had at least 9 opsins, they found that none of the lineages that evolved from the urbilaterian retained all 9 opsins.

Since gains and losses are common among opsins, we did not reconcile our phylogeny with a species tree, as Ramirez et al. [35] did for their phylogeny. Since they are common, the number of chromopsin paralogues may indeed range from one to seven in the urbilaterian.

### 4.2. The Function of the Chromopsins

The chromopsins are an interesting group of opsins that have diverse and poorly understood functions. Some chromopsins have a derived NPxxY^7.53^ motif (Figure 3), which may change their G-protein interaction or activation. Furthermore, many chromopsins bind all-trans-retinal in the dark, including peropsins [91], varropsins [74,92], RGR-opsins [93], and retinochromes [69]. This is unusual as most opsins bind 11-cis-retinal in the dark and convert it to all-trans-retinal when illuminated. Among the chromopsins, the gluopsins lack the well conserved lysine 296^7.43^ (Figure 2C and Figure 3). This may prevent them from binding retinal and raises fascinating questions about their evolution and function. The gluopsins could function like other opsins as thermoreceptors [30] or be involved in mechanoreception such as hearing [31].

However, since the gluopsins are more closely related to the other chromopsins, these may give us some clues about their function. The best-studied chromopsins are retinochromes and RGR-opsins. They have, like the gluopsins, a derived NPxxY^7.53^ motif (Figure 3), and thus are claimed not to signal [80], but to produce 11-cis-retinal as photoisomerases. This view is considered established in the literature [33,74,80,94,95]. Therefore, it would be reasonable to assume that the gluopsins might be photoisomerases as well. Then, the missing lysine would be an optimization in a high throughput system as covalent binding might cost time. However, we disagree with the literature that the retinochromes and RGR-opsins are established as photoisomerases.

Only if RGR-opsins and retinochromes are indeed photoisomerases, it is reasonable to assume that the gluopsins are photoisomerases. Therefore, we discuss what a photoisomerase is, how something could be experimentally shown to be a photoisomerase, and how a separate photoisomerase could be useful. Then, we evaluate what is known about the different chromopsins beyond their existence, and finally we discuss possible other functions.

### 4.3. Photoisomerases

A photoisomerase in the general sense is an enzyme that converts a molecule from one isomer to another with the energy of a photon. For that, it binds the molecule before the reaction and releases it afterwards as a photoproduct. Whether the photoproduct is just a byproduct, or the main product used for something else does not matter in this definition. However, under this definition, cattle rhodopsin, and likely most cilopsins, are photoisomerases, since they release all-trans-retinal once they have converted it from 11-cis-retinal [17]. Here, all-trans-retinal is considered as a byproduct, while the main function is the phototransduction cascade, which is activated by the photoisomerization. In contrast, retinochromes and RGR-opsins have been claimed to produce 11-cis-retinal for other opsins only, and not to activate phototransduction cascades at all [80,81].

For retinal, photoisomerases change the absorption spectrum: 11-cis-retinal absorbs maximally at 380 nm [7] and all-trans-retinal at 387 nm [17]. In contrast, 11-cis-retinal covalently bound to cattle rhodopsin absorbs maximally at 498 nm [96]; and all-trans-retinal covalently bound to RGR-opsin pH-dependently at 469 nm or 370 nm [97], and to retinochrome at 492 nm [68]. Furthermore, binding all-trans-retinal to a lipid in the plasma membrane moves the absorption maximum to 450 nm, and thus this system acts as a protein free photoisomerase [98]. Another kind of photoisomerase exists in the honeybee. It is water-soluble and thus not a transmembrane domain protein such as opsins [99,100].

To show that something is a photoisomerase in the general sense is relatively easy. To show that the photoproduct is used and needed for something else is more difficult. Ideally, if the photoproduct is missing, a phenotype should result. Even if the photoisomerization only supplies a small fraction (e.g., 10%) of what is needed, it still should result in a phenotype, if missing, so that a function can be established.

To make photoisomerases work effectively, they should supply a significant amount of retinal to their target opsin. To do that they need to be present in a comparable amount or in a much higher amount as their target opsins. Additionally, they also should be placed next to their target opsins so that the distance for exchanging retinal is short. This requires the photoisomerase to use as much space as the target opsin. This might be a problem in rods, where the membranes are stacked to hold as many target opsin molecules as possible to catch every photon under low light, scotopic conditions.

These photons can either be used for vision or for the photoisomerase, unless the photoisomerase contributes to visual excitation, as well. This could be either achieved by binding to a different G-protein or by modifying the binding pocket of the photoisomerase, so that the pocket then binds all-trans-retinal in the dark and converts it to 11-cis-retinal for signaling. Basically, the target opsin would also become a photoisomerase in a two-opsin system, where both opsins are photoisomerases for each other. In this system, both opsins can be tuned spectrally independently of each other.

This might be not possible in a system with bistable opsins. Bistable opsins can convert all-trans-retinal back to 11-cis-retinal by absorbing another photon of a different wavelength without releasing retinal [80,101]. These bistable opsins would integrate the function of a photoreceptor, converting 11-cis-retinal to all-trans-retinal to activate the receptor and to convert it back to supply the receptor with 11-cis-retinal. However, the wavelength sensitivities of both photoreactions may depend on each other because they depend on the same molecule.

Having two opsins functioning as photoisomerase for each other or using bistable opsins might solve the space problem. However, the space problem is only crucial for high performance tasks such as vision or the UV-avoidance response of the larva of *Platynereis dumerilii,* which is also fast [102] and uses photoreceptors with stacked membranes [103]. In tasks that do not require the detection of many photons during a short period, photoisomerases might be more useful, but even here the photoisomerase and its target opsin should be next to each other.

### 4.4. The Varropsins

The varropsins also bind all-trans-retinal in the dark state and isomerase it to 11-cis-retinal upon light exposure [74,92]. This way they are dark-active opsins that are deactivated by light [92]. Varropsins are expressed in the eyes of the spiders *Hasarius adansoni* [74] and *Cupiennius salei* [104]. In Limulus, a varropsin is expressed in glia and pigment cells of the eyes next to photoreceptor cells and in the central nervous system [75,88].

### 4.5. The Peropsins

In mice, a peropsin is localized to the apical microvilli of the retinal pigment epithelium (RPE) [72]. There, it regulates storage or the movement of vitamin A from the retina to the RPE [105]. A peropsin is also expressed in keratinocytes of the human skin. In keratinocyte cell culture, it reacts to UV light if retinal is supplied [106]. In chicken, a peropsin and an RGR-opsin are expressed in the pineal gland and the retina [107]. In amphioxus, a peropsin binds in the dark-state all-trans-retinal instead of 11-cis-retinal [91]. Despite peropsins having been discovered 25 years ago, in 1997 [72], not much more is known about them. This might be, because the human peropsin could not be linked to an eye disease [108,109], which contrasts with human RGR-opsin, which could be linked to retinitis pigmentosa [110].

### 4.6. The RGR-Opsins

The RGR-opsins have an NAxxY^7.53^ motif, instead of the well-conserved NPxxY^7.53^ motif (Figure 3, outgroup). This motif is important for G-protein binding. For instance, if it is mutated to NAxxY^7.53^ in the rat m3 muscarinic receptor, the receptor can still be activated but less efficiently [20]. Therefore, RGR-opsins are thought to neither signal nor activate a phototransduction cascade [80,81].

However, the human MT2 melatonin receptor signals via a G-protein and has an NAxxY^7.53^ motif natively. If this motif is mutated to NPxxY^7.53^, the receptor cannot be activated, but can be rescued partially if the motif is mutated to NVxxY^7.53^ [22]. Furthermore, when the motif is mutated to NAxxY^7.53^ in cattle rhodopsin, the mutant has 141% of wild type activity [19]. This evidence shows that a GPCR does not need a standard NPxxY^7.53^ motif to signal.

RGR-opsins bind all-trans-retinal instead of 11-cis-retinal in the dark [93] and are involved in retinal regeneration [111]. Therefore, RGR-opsins were thought to be photoisomerases [81]. However, in the retinal pigment epithelium (RPE) cells, they are located in the smooth endoplasmic reticulum [112] and regulate retinoid traffic and production [113,114]. In particular, they speed up the production of 11-cis-retinol (an 11-cis-retinal precursor) from all-trans-retinyl-esters, light-independently [115]. The all-trans-retinyl-esters, however, are made available light-dependently by the same RGR-opsins for the Rpe65-isomerase in the RPE. Therefore, RGR-opsins signal, but it is unclear whether they signal via a G-protein or some other mechanism [116]. This is contrasted by the results of Zhang et al. [117], who found that the Rpe65-isomerase activity does not depend on light. However, they used a cell-free RPE-microsome system for their experiments. Microsomes are generated from cell fragments, and thus lack internal lipid storage where substrate for the Rpe65-isomerase could come from. Basically, this only shows that RGR-opsins are photoisomerases in the general sense, it does not show whether they also produce all-trans-retinal for other opsins.

Although, RGR-opsins are present in a relatively high amount compared to the total amount of protein in RPE-cells [70], RPE-cells do not stack their membranes as densely as rods and cones do, so the amount of RGR-opsin in RPE-cells should be relatively low compared to visual opsins in rod and cones. Additionally, RGR-opsins have in vitro only 34% of the photosensitivity of cattle rhodopsin, and they do not readily release 11-cis-retinal, which can be displaced by all-trans-retinal [93]. Finally, RGR-opsins are located in the RPE behind the rod and cones, which take out a significant fraction of photons. In sum, all these properties do not support RGR-opsins as photoisomerases for visual opsins.

Therefore, RGR-opsins apparently function in the RPE primarily as photoreceptors. They may just work in reverse: In the dark, they are active and are inactivated by light, like chicken Opn5L1 (Opn8), which is inactivated by light by isomerizing all-trans-retinal to 11-cis-retinal [118].

In principle, RGR-opsins could still contribute to chromophore production in the RPE as photoisomerases. However, this is difficult to determine, since they increase the substrate supply for another isomerase [116], and an RGR-opsin knockout thus removes both the 11-cis-retinal produced by the RGR-opsin itself and that from the isomerase.

The cone outer segments also contain RGR-d, an RGR-opsin splice variant [119]. However, RGR-d lacks most of transmembrane domain 6 and thus is inactive [117], but it could still indicate RGR-opsin expression [119]. This could fulfill the need for a highly efficient and abundant photoisomerase in the cone disks next to their target opsins. Additionally, in vitro experiments suggest that RGR-opsins may serve as photoisomerases in Müller cells [120] where they are located in the endoplasmic reticulum [112]. However, whether RGR-opsins serve as photoisomerases in cones or whether they contribute a significant amount to chromophore production on their own in the RPE, remains to be seen. It rather seems that the RGR-opsins are provided with all-trans-retinal by the visual opsins and thus the visual opsins serve them as photoisomerases.

Furthermore, if RGR-opsin was present in a high amount then it would reflect so much light of specific wavelengths to the eyes of an observer that it would act as a pigment and give color to its host cells, like rhodopsin, which stains the rods purple. This would be visible to curious researchers; however, the first RGR-opsins were not discovered visually but rather via molecular techniques [70,71]. In fact, 28 years before the first RGR-opsin was found, the first chromopsin was discovered by eye [68]. It was subsequently named retinochrome [69].

### 4.7. The Retinochromes

Like RGR-opsins, retinochromes are also thought to function as photoisomerases. They bind in the dark state all-trans-retinal [121], which is isomerized by light to 11-cis-retinal and released immediately, so that the retinochrome can readily take up another molecule of all-trans-retinal. Squid retinochrome is so abundant that it could be discovered by eye [68]. In an in vitro solution, it can supply cattle rhodopsin with 11-cis-retinal, regenerates with all-trans-retinal several tens of times faster than cattle rhodopsin with 11-cis-retinal [122], and it is more light-sensitive than the visual squid opsin [121]. Therefore, retinochrome would serve as an effective photoisomerase. However, in the squid eye, it is mostly located in the inner segments of the photoreceptor cells, while the visual opsin is located in the outer segments [123]. The inner and outer segments are separated by a dense screen of pigment, so that retinochrome could only use light that comes through body tissues and not from the eyes [68]. Some retinochrome is also found in the outer segments, but in lower amount and not next to the visual opsins in the rhabdoms [124]. However, the visual opsins and the retinochromes could exchange retinal via a shuttle protein [125].

In contrast to the squid eye, the visual opsin is co-expressed with retinochrome in extraocular tissue, such as the longitudinal bundles of central fin muscle, the arm ganglia, the sucker peduncle nerves, the epidermal hair cells, and the parolfactory vesicles [126]. In the parolfactory vesicles, the amount of visual opsin and retinochrome is roughly equal [127], and thus retinochrome could serve there as a photoisomerase for the visual opsin.

This evidence shows that retinochromes can function as photoisomerases. However, this comes from biochemical in vitro and in situ experiments, which are highly artificial systems, and thus cannot tell whether retinochromes also function as photoisomerases in vivo.

### 4.8. The Gluopsins: Opsins without Lysine 296^7.43^

Here, we have covered everything that is known about the chromopsins and their functions. From that, we conclude that the evidence is not enough to determine if RGR-opsins and retinochromes are photoisomerases. This influences the plausibility whether gluopsins are photoisomerases. Therefore, we should also think beyond the photoisomerase hypothesis.

Recently, the *Drosophila* rhabopsins Rh1, Rh4, and Rh7 were reported to function not only as photoreceptors, but also as chemoreceptors for aristolochic acid. These opsins still have lysine 296^7.43^ like other opsins. However, if this lysine is replaced by an arginine in Rh1, then Rh1 loses light sensitivity but still responds to aristolochic acid. Thus, lysine 296^7.43^ is not needed for Rh1 to function as chemoreceptor [26]. Therefore, we wondered whether any opsins existed that had lost lysine 296^7.43^ during evolution. Such opsins would form a clade embedded within the other opsins.

Indeed, we found such a clade: the gluopsins. The gluopsins have glutamic acid instead of lysine 296^7.43^ and form a strong clade of 36 member sequences (Figure 2C and Figure 3). We also reconstructed the astropsins with one member that also has glutamic acid 296^7.43^ (Figure 2C and Figure 3). Previously, three members of the astropsins have been reported to have glutamic acid 296^7.43^. These members were mined from transcriptome and genome databases [37]. However, we could trace two astropsin sequences with glutamic acid 296^7.43^ to two independent genome projects from sea urchins [128,129]. Beyond their sequences, however, nothing is known about them. Beside the gluopsins and astropsins, the nemopsins have lysine 296^7.43^ replaced, however with arginine (Figure 2C and Figure 3), they also have a conserved NPxxY^7.53^ motif. A nemopsin is expressed in chemosensory cells in *C. elegans*. Therefore, the nemopsins are thought to be chemoreceptors [77].

All other opsins without lysine 296^7.43^ are isolated single sequences, so that we can assume they have been missequenced, misassembled, or are pseudogenes. For example, in an encephalopsin of the clawed frog, *Xenopus tropicalis* (XM_002935666), lysine 296^7.43^ is replaced by isoleucine and thus is regarded by Kato et al. [95] as a pseudogene. It has been subsequently removed from the NCBI database. Therefore, we do not consider single opsins without lysine 296^7.43^ within a clade as real, unless these opsins are at least sequence verified and ideally characterized functionally or by expression.

Gluopsins have been previously described in butterflies, dragonflies, scorpionflies, and in beetles. They have been confirmed in dragonflies by expression data and their sequences have been verified by Sanger sequencing of their cDNA [78]. They were also found in head transcriptomes of scorpionflies [79]. They have been reconstructed as a clade by Henze and Oakley [73], who however did not mention glutamic acid 296^7.43^. Beyond that, nothing is known about them, especially about their function.

The gluopsins share with RGR-opsins and the retinochromes a derived NPxxY^7.53^ motif. The motif in gluopsins is PVxxY^7.53^ or PLxxY^7.53^ (Figure 3). This has two mutations compared to the NAxxY^7.53^ motif of the RGR-opsins. Even with this derived motif, we should not exclude that gluopsins signal unless shown otherwise experimentally, since the whole receptor could have acquired compensatory mutations.

Indeed, gluopsins should signal, otherwise they could not function as chemoreceptors, which is a possibility. They could sense chemicals like the *Drosophila* opsins Rh1, Rh4, and Rh7, which sense aristolochic acid without lysine 296^7.43^ [26]. However, in cattle rhodopsin, the retinal binding lysine can be shifted from position 296 to other positions, even into other transmembrane domains, without changing the activity [130]. This way the gluopsins could serve as photoreceptors or even as photoisomerases. Even so, it has not been conclusively shown that RGR-opsins and retinochromes are photoisomerases as we have discussed above.

From inspecting our alignment manually, we could not find an alternative position for the retinal binding lysine that is conserved across all gluopsins. However, different gluopsins may have switched the retinal binding lysine to different positions so that all the gluopsins could serve as photoreceptors. Beside light and chemicals, the gluopsins could, like other opsins, also detect heat, phospholipids, mechanical stimuli, or other stimuli [32,33]. In the end however, whether the gluopsins are photoreceptors, chemoreceptors, or something else remains to be determined experimentally.

## 5. Summary

To answer our question, whether opsins exist that have lost lysine 296^7.43^ during evolution, we built an automatic phylogeny pipeline that can be easily adjusted for reconstructing the phylogeny of other GPCRs and other proteins. We reconstructed the first 5k opsin phylogeny, which contains more than 5 times the number of opsin sequences compared to previous large-scale phylogenies [28,35]. The full description of this phylogeny will be published elsewhere. Finally, we answered our question: Opsins that lost lysine 296^7.43^ during evolution do exist. In these opsins, lysine 296^7.43^ is replaced by glutamic acid 296^7.43^, and thus we call them gluopsins. The gluopsins are found in insects such as beetles, scorpionflies, dragonflies, and butterflies including the silk moth, which is of commercial interest. The gluopsins are an exciting target to study the fascinating functional flexibility of opsins, especially as more opsins with functions beyond light sensitivity are discovered. However, what the function of the gluopsins is, is unknown and so it is to be answered by future research.

## Figures and Tables

**Figure 1 cells-11-02441-f001:**
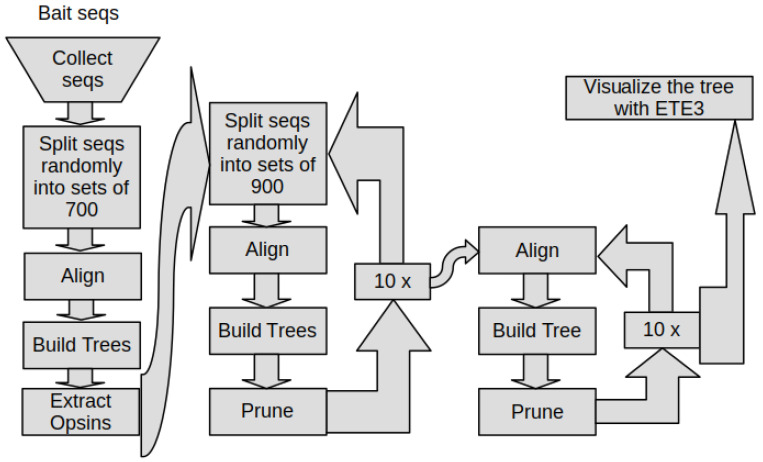
Flowchart of the pipeline used for reconstructing our opsin phylogeny. The pipeline starts with the bait sequences and uses them to collect similar sequences with BLAST from public sequences databases. The sequences are filtered for opsins by building phylogenetic trees and extracting the opsin clade from them. It adds back outgroup sequences (not shown in the chart). With this set, it then reconstructs small trees ten times for fast rogue pruning, followed by trees with the full sequence set for rogue pruning ten times, since what a rogue is, is determined by all other sequences in the set. After rogue pruning, the final tree is visualized with ETE3. The last step is manual and can be applied to all trees.

**Figure 2 cells-11-02441-f002:**
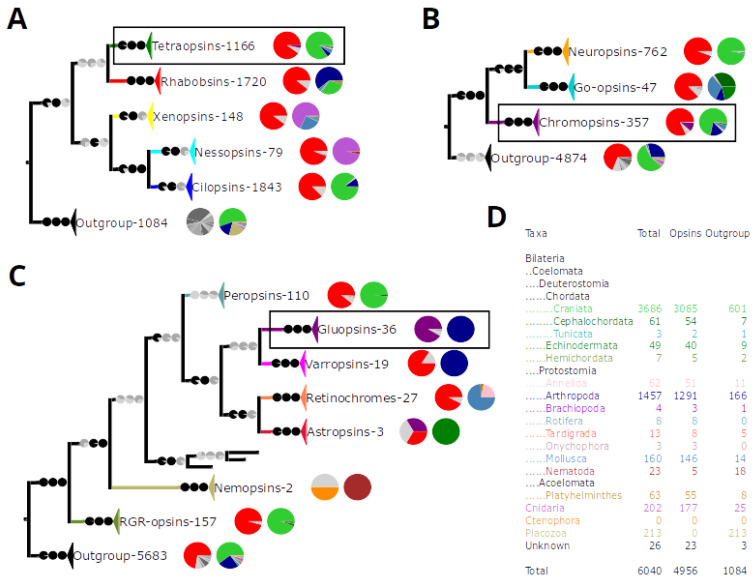
Phylogenetic reconstruction of the opsins. (**A**) The groups of the opsins and the outgroup are collapsed. The outgroup contains all other GPCRs. The frame highlights the tetraopsins, which are expanded in (**B**). (**B**) The groups of the tetraopsins are shown. The outgroup contains all other opsins and the non-opsin GPCRs. The frame highlights the chromopsins, which are expanded in (**C**). (**C**) The groups of the chromopsins are shown. The frame highlights the gluopsins. The outgroup contains all other opsins and the non-opsin GPCRs. (**A**–**C**) Next to each clade is the number of sequences within that clade shown. The first pie chart shows the percentage of a certain amino acid at lysine 296^7.43^. Red stands for lysine (K), purple stands for glutamic acid (E), the other amino acids are alternatively colored dark or mid-gray so that two adjacent amino acids have different shades of gray in the pie chart. Light gray stands for a gap at this position. This comes from sequences that are incomplete and do not contain the seventh transmembrane domain. The second pie chart gives the taxon composition for each clade, the colors correspond to the list of taxa in (**D**) The taxon composition is also given in a numerical format in Appendix A. The support values are given as pie charts. They are from right to left SH-aLRT/aBayes/UFBoot. Splits are considered supported when SH-aLRT ≥ 80%, aBayes ≥ 0.95, and UFBoot ≥ 95%. If a support value is above its threshold the pie chart is black otherwise gray. (**D**) The list of higher taxa represented by the sequences. A few sequences are unidentified, since their sequence identifiers do not contain a genus name.

**Figure 3 cells-11-02441-f003:**
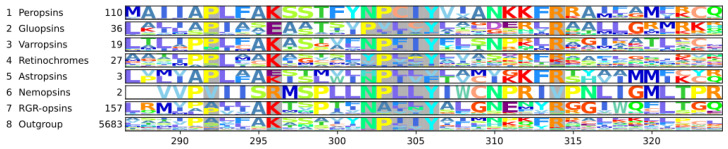
Consensus sequences of the different chromopsins. The first column contains a number for each chromopsin group for easy reference. The second column shows the names for each group. The groups are in the same order as in the tree in Figure 2C. The third contains the number of sequences in each group. And the fourth column contains the sequence logo, the height of the letters indicates the percentage of that amino acid given at that position. The x-axis gives the position of the amino acid corresponding to cattle rhodopsin. Positions 292^7.39^ and 314^7.64^ are highlighted in gray. Lysine (K) 296^7.43^ is highlighted with a gray background, which is replaced in the nemopsins by arginine (R) and in the gluopsins by glutamic acid (E). The NPxxY^7.53^ motif is highlighted with a gray background.

## Data Availability

The code of the phylogeny pipeline is available at https://github.com/MartinGuehmann/PhylogenyPipeline (accessed on 1 June 2022). The final sequences, the alignments, and the newick tree for the final iteration are included in the Appendix A. The complete data generated by the pipeline is available at https://github.com/MartinGuehmann/Opsins (accessed on 1 June 2022). Our version of PIA is available at https://github.com/MartinGuehmann/PIA2 (accessed on 1 June 2022).

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
