# Peer review of "The Gluopsins: Opsins without the Retinal Binding Lysine"

_cells, 2022, doi:10.3390/cells11152441_

Round 1

Reviewer 1 Report

This is a solid study that explores the phylogeny of a "new" class of biologically active molecules which they name "gluopsins."   This is interesting work that will be referenced in years ahead.   The study is sufficiently complete in current form.  

Author Response

We thank you for reviewing our work and are pleased that you like it. “This is interesting work that will be referenced in years ahead.”, is a sentence to cite and so well crafted.

Reviewer 2 Report

The work of Gühmann et al. aim to provide novel insights into the evolution of opsins and their possible function when they are not able to catch a photon and thus transmit light perception. They were able to define a clade of 33 opsin-related proteins, from 4956 opsin-related proteins in total, where the retinol binding lysine had been replaced by a glutamic acid, which they named gluopsins. Though the amount of novel data above what has been published before by several other laboratory is limited, their data is consistent and of very high quality.

In general, the presentation of the data is sound and is of interest for the readership of Cells after minor revision.

Minor points:

In the center of their work is the identification of what they call the gluopsins. However, the phylogenetic tree in the supplement is by far to large to be useful for the reader. It is almost impossible to check for the gluopsins. I suggest preparing a much smaller version where the gluopsins clade becomes clear. Even a sequence alignment of the gluopsins may be of interest, which indicated exon patterns.

Please define what is a rogue sequence in your opinion. Just to take out sequences from the alignment that do not fit (but are real, expressed) may end in a bias within the tree. Moreover, I not sure why incomplete sequences have been included, or are these opsin-like proteins that do miss one or more of the transmembrane domains?

Figure 1:

The color code of the second pie is almost impossible to read. Please, add a table in the supplementary.

The numbers in c do not count to 357 of the chromopsins. What are the orphan branches?

Line 326: the 296 7.43 is not superscript

Line 380: … and since we them from…: verb is missing, sentence incomplete

Discussion:

The discussion is pretty extended and not really focused on their question raised and thus should be shortened. Many issues are not important to understand the work described and the conclusions, example the photoisomerases and their absorption spectra. Some issues are highly speculative with no relation to the new gluopsins and should be taken out.

Author Response

We thank you for your positive and helpful comments. We are especially thankful for the comment that made us to include the nemopsins, as this is something we do not like to miss. We reply to each comment below and explain the revisions we made.

Point 1: In the center of their work is the identification of what they call the gluopsins. However, the phylogenetic tree in the supplement is by far to large to be useful for the reader. It is almost impossible to check for the gluopsins. I suggest preparing a much smaller version where the gluopsins clade becomes clear.

Response 1: We added a version of the tree that only shows the Chromopsins expanded, and all the other clades collapses as outgroup, as you have requested. The supplement also includes the tree in newick format, which can be visualized in fine detail as the reader prefers.

Point 2: Even a sequence alignment of the gluopsins may be of interest, which indicated exon patterns.

Response 2: We agree that the intron and exon patterns would be interesting in their own right, as they could clarify how the gluopsins are related to the other chromopsins. However, within the scope of this analysis (to demonstrate the existence of putatively-functional opsins without the chromophore-binding lysine), and considering the journal’s short turnaround for revisions, this additional analysis is not practical at this time.

Point 3: Please define what is a rogue sequence in your opinion. Just to take out sequences from the alignment that do not fit (but are real, expressed) may end in a bias within the tree.

Response 3: When Aberer et al. (2013) described RogueNaRok, which we use for removing rogue sequences, they define rogues as:

“The resolution in a consensus tree and the branch support on the best-known tree can be substantially deteriorated by rogues (the term rogue/rogue taxa was introduced by Wilkinson 1996), which assume varying and often contradictory positions in the tree set.”

Saunders et al. (2019) introduce rouge sequences the following:

“A fairly large number of different researchers (Ashlock et al., 2009; Aberer and Stamatakis, 2011; Ting et al., 2014; Saunders, 2017; Saunders et al., 2018; Westover et al., 2013) have detected that hierarchical clustering algorithms suffer from a type of instability in which adding or deleting even a single taxon from a data set can cause a substantial rearrangement of the resulting tree produced by the algorithm.”

In section 2.3 we have clarified our definition and added references to Aberer et al. and Saunders et al. so it reads now:

“Rogue sequences introduce instability to a phylogeny by jumping around from one place to another, and may change the relationships and branch supports within the phylogeny unpredictably. (Aberer et al. 2013, Saunders et al. 2019)”

Point 4: Moreover, I not sure why incomplete sequences have been included, or are these opsin-like proteins that do miss one or more of the transmembrane domains?

Response 4: Previous large opsin phylogenies included such sequences, (Ramirez et al. 2016, D’Aniello et al. 2015, Böhm et al. 2018), and we wanted to sample as many opsins as possible to build substantially on these previous efforts. If we manually inspect our sorted alignment, then we see in general and subjectively that the fragments are in the right place.

The general assumption is that no opsins exist that lack one or more of the transmembrane domains and are still functional. However, this is still an interesting hypothesis that should be tested in future work. If such opsins existed, they should be conserved and form a clade. However, we do not see something like this in our sorted alignment.

Figure 1:

Point 5: The color code of the second pie is almost impossible to read. Please, add a table in the supplementary.

We added such a table to the supplement.

Response 5: The numbers in c do not count to 357 of the chromopsins.

The numbers add up to 352, plus the 5 orphan branches gives 357.

Point 6: What are the orphan branches?

Response 6: That is a good question. There orphan branches consist of two groups, one consists of two sequences from mollusks and one from an annelid. In some of our trees, these sequences group with the peropsins, so they could indeed be peropsins. However, since they do not group in our final tree with the peropsins, this is questionable. Also, the internal nodes of this group are not supported according to our threshold. Therefore, we prefer not to assign them to a group name and not to address them in the manuscript.

The other two sequences are nematode sequences. These are the only sequences left of this group in our final tree. In trees of previous rounds, this groups contains more sequences including sequences from Caenorhabditis elegans (NP_001364737.1) and Pristionchus pacificus (PDM61246.1). These sequences have lysine replaced by arginine. This way, we could call them argopsins. However, argopsin is already the name of a chemical compound, so we call them nemopsins since they are nematode specific. They have been described before including in some expression studies, so they appear to be real. However, we still would like to focus on the gluopsins and just added a results section for them and mention them briefly in the discussion.

We updated Figure 2 to include the nemopsins and added the following new results section:

“3.4. The nemopsins have arginine at position 296

The nemopsins are nematode chromopsins. Only two nemopsins remain in our final tree, and only one covers the 7th transmembrane domain where lysine 2967.43 is replaced by an arginine (Figure 2c, Figure 3, Supplementary Figure S1). Among the removed sequences are sequences from Caenorhabditis elegans (NP_001364737.1) and Pristionchus pacificus (PDM61246.1); both species are model systems and both sequences have the arginine. The sequence from C. elegans has been previously described as an opsin like GPCR with arginine 2967.43 and a conserved NPxxY7.53 motif. It was named sro-1, short for serpentine receptor class o 1 and is expressed in chemosensory cells [77] The nemopsins have not been included in a previous opsin phylogeny, and we are not aware of anything else known about them.”

And added the following to the final discussion section:

“Beside the gluopsins and astropsins, the nemopsins have lysine 2967.43 replaced, however with arginine (Figure 2c, Figure 3), they also have a conserved NPxxY7.53 motif. A nemopsin is expressed in chemosensory cells in C. elegans. Therefore, the nemopsins are thought to be chemoreceptors [77].”

Point 7: Line 326: the 296 7.43 is not superscript

Response 7: Fixed.

Point 8: Line 380: … and since we them from…: verb is missing, sentence incomplete

Response 8: Fixed. The sentence reads now: “However, since we have 33 gluopsins with glutamic acid, and since we have recovered them from different higher insect taxa, we can exclude that those are missequenced, misassembled or pseudogenes.”

Discussion:

Point 9: The discussion is pretty extended and not really focused on their question raised and thus should be shortened. Many issues are not important to understand the work described and the conclusions, example the photoisomerases and their absorption spectra. Some issues are highly speculative with no relation to the new gluopsins and should be taken out.

Response 9: We agree the discussion is extensive, but we feel that its current form is important for understanding the gluopsins in the context of opsin evolution and function. The gluopsins form a group with RGR-opsins, retinochromes, and peropsins. Like RGR-opsins and retinochromes, the gluopsins have a derived NPxxY motif suggesting that gluopsins retain the potential to signal despite their absent retinal binding lysine. Therefor it is important to consider the potential function(s) of the gluopsins in the context of their related opsin clades.

One possibility is that they are photoisomerases like RGR-opsins and retinochromes. The literature often definitively describes opsins from these clades as such:

“In addition, there is a third subfamily called Goopsins (coupled to Go proteins) and a more heterogeneous group including retinochrome, peropsin, and neuropsin which serve as photoisomerases.” (Gehring 2014)

“This group called RGR/Go-opsins (also referred to as Group 4 opsins) includes retinal G protein-coupled receptor opsins (RGR), peropsin, and neuropsin (Opn5) (Figure 3, Tables 2–4). These opsins function as photoisomerases or couple to different signaling cascades (Figure 2c, Tables 2– 3).” (Leung and Montell, 2017)

“There are seven major opsin subfamilies in chordates: melanopsin (opsin 4); encephalopsin/panopsin and teleost multiple tissue (TMT) opsin (opsin 3); ciliary photoreceptor opsins including rod/cone opsins, pinopsin, and vertebrate-ancient (VA) opsin; Go-coupled opsins; opsin 5 (formerly neuropsin); peropsin; and photoisomerases.” (Kato et al. 2016) With photoisomerases Kato et al. mean the retinochromes and RGR-opsins.

“Retinochrome and RGR, the members of the retinal-photoisomerase subfamily, bind all-trans retinal (Figure 2b) as a chromophore [67,68] and are not coupled to G proteins, unlike the visual opsins, which bind the 11-cis form of retinal.” (Terakita 2005)

“In addition, peropsin contains an ‘NPxxY’ amino acid sequence motif, which is highly conserved among most opsins and other GPCRs but not in photoisomerase retinochrome and RGR, implying that peropsin might drive G protein-mediated signaling.” (Tsukamoto and Terakita)

These previous studies and reviews imply with little doubt that RGR-opsins and retinochromes are photoisomerases, and imply that because retinochromes and RGR-opsins do not have the NPxxY motif, that they do not signal. However, since the gluopsins have the derived NPxxY motif and are more closely related to the retinochromes and RGR-opsins, it is reasonable to assume that the gluopsins are photoisomerases as well. We feel that it is crucial to caution against this interpretation and open lines of inquiry into alternate hypotheses for the function of gluopsins and other poorly understood clades.

Therefore, we discuss what actually is a photoisomerase, how to experimentally show that something is a photoisomerase, and what an additional photoisomerase may be good for. Then we review what is known about the other chromopsins. Though there is little previous information regarding the peropsins and the varropsins, while retinochromes and the RGR-opsins have received some more attention. In case of the RGR-opsins, we discuss whether the mutated NAxxY motif could prevent them from signaling. The current evidence says that this is not the case, making them less likely to be photoisomerases. We then discuss what else is known about the RGR-opsins and retinochromes, and we cover here everything that is known and relevant about whether they could be photoisomerases. We conclude that there is currently not enough evidence to determine if RGR-opsins and retinochromes are exclusively photoisomerases, and that further research is needed.

We hope that our discussion gives researchers interested in retinochromes and RGR-opsins new ideas and drives interesting new studies. Furthermore, we also hope to inspire people who want to work on the gluopsins to think beyond the photoisomerase hypothesis. Therefore, our discussion might be extended, but we think it is valuable for the opsin community. And therefore, we did not shorten it. However, we clarified our rational for these sections according to the above outline and inserted two paragraphs at the end of section 4.2 before we define what a photoisomerase is:

“However, since the gluopsins are more closely related to the other chromopsins, these may give us some clues about their function. The best-studied chromopsins are retinochromes and RGR-opsins. They have, like the gluopsins, a derived NPxxY7.53 motif (Figure 3), and thus are claimed not to signal [80], but to produce 11-cis-retinal as photoisomerases. This view is considered established in the literature [33,74,80,94,95]. Therefore, it would be reasonable to assume that the gluopsins might be photoisomerases as well. And the missing lysine would then be an optimization in a high throughput system as covalent binding might cost time. However, we disagree with the literature that the retinochromes and RGR-opsins are established as photoisomerases.

Only if RGR-opsins and retinochromes are indeed photoisomerases, it is reasonable to assume that the gluopins are photoisomerases. Therefore, we discuss what a photoisomerase is. How something could be experimentally shown to be a photoisomerase. And how a separate photoisomerase could be useful. Then we evaluate what is known about the different chromopsins beyond their existence, and finally we discuss possible other functions.”

And at the beginning of section 4.8 we sum up our discussion of the photoisomerases and the other chromopsins:

“Here, we cover everything that is known about the chromopsins and their functions. From that, we conclude that the evidence is not enough to determine if RGR-opsins and retinochromes are photoisomerases. This influences the plausibility, whether gluopsins are photoisomerases. Therefore, we should also think beyond the photoisomerase hypothesis.”

Reviewer 3 Report

I find the article very interesting indeed. The authors constructed a pipeline of the large-scale opsins to reconstruct their phylogeny.  Among them they found opsins, called glucopsins, which have lysine replaced by glutamic acid whose function has not yet been determined. The authors have done a nice thorough job and highlighted the role of evolution on some well-conserved proteins, opening up new insights and new phylogenetic perspectives

Author Response

We thank you for your review and are pleased that you like our work.